# Mechanical ventilation in acute myocardial infarction: Outcomes from a prospective audit at a cardiovascular hospital in Pakistan

**Muhammad Imran Ansari**[ORCID]*, **Madiha Umair, Lalarukh Taimoor, Aziz-ur Rehman Memon, Zohaib Abubaker, Muhammad Sohaib Arif, Nadeem Qamar, Musa Karim, Jawed Abubaker**

National Institute of Cardiovascular Diseases (NICVD), Karachi, Pakistan

* dr.mimran@hotmail.com

**Data Availability Statement:** All relevant data are within the paper and its Supporting Information files.

## Abstract

### Background

This clinical audit aims to evaluate the clinical data regarding the management and outcomes of acute myocardial infarction (AMI) patients requiring mechanical ventilator (MV) support, along with identifying factors associated with prolonged MV support and 180-day mortality.

### Materials and methods

In this study, we audited clinical data regarding management, in-hospital and short-term outcomes of adult patients with AMI required MV support. Patients with prolonged MV duration (>24h) and/or 180-day mortality were compared with their counterparts, and associated factors were identified. The binary logistic and Cox regression analyses were performed to determine the predictors of prolonged MV duration and 180-day mortality.

### Results

In a sample of 312 patients, 72.8% were male, and the mean age was 60.3±11.5 years. The median MV duration was 24 [24–48] hours, with 48.7% prolonged MV duration. The admission albumin level was found to be the independent predictor of prolonged MV duration with an adjusted OR of 0.42 [0.22–0.82]. Overall 7.4% were re-intubated, 6.7% needed renal replacement therapy, 17.6% required intra-aortic balloon pump (IABP) placement, and 16.7% required temporary pacemaker placement. The survival rate was 80.4% at the time of hospital discharge, 74.7% at 30-day, 71.2% at 90-day, and 68.6% at 180-day follow-up. Age, prolonged MV duration, and ejection fraction were found to be the independent predictors of cumulative 180-day mortality with adjusted HR of 1.04 [1.02–1.07], 1.02 [1.01–1.03], and 0.95 [0.92–0.98], respectively.

### Conclusions

Prolonged ventilator duration has significant prognostic implications; hence, tailored early recognition of high-risk patients needing more aggressive care can improve the outcomes.

**Funding:** The author(s) received no specific funding for this work.

**Competing interests:** The authors have declared that no competing interests exist.

## Introduction

Cardiovascular diseases, including acute myocardial infarction (AMI), are the leading cause of mortality and morbidity [1]. With increasing life expectancy, percutaneous coronary intervention (PCI) is being performed more frequently. PCI contributes to a better outcome in AMI [2]. Despite timely reperfusion therapies, complications from AMI, such as arrhythmias and cardiogenic shock, often necessitate admission to intensive care units (ICU) [3]. In AMI, complicating with cardiogenic shock requiring advanced life support, such as mechanical ventilation, vasoactive therapies, and mechanical devices like intra-aortic balloon pump (IABP) or pacemaker devices have been associated with mortality and morbidity [3, 4]. In addition to increased risk of mortality and morbidity, ICU admission is also associated with an increased cost of care, with an up to 2.5 times higher estimated cost for ICU patients when compared to non-ICU patients [5, 6]. The increased cost of care has implications for the resource-limited healthcare system of low and lower-middle-income countries, such as Pakistan.

The National Institute of Cardiovascular Diseases (NICVD) and its satellite centers are Pakistan's largest cardiac care network, providing free-of-charge services since 2015 with the support of the government of the province of Sindh. The Karachi center has a dedicated independent critical-care medicine service to provide sub-specialty support with a 20-bed capacity for ventilators. On average, the center sees and manages 100 post-PCI mechanically ventilated patients on advanced life support per month. With the addition of multiple sub-specialty ICU services, the quality of care of these patients has improved significantly, and a reduction in mortality and morbidity has been observed.

This clinical audit aims to evaluate the clinical data regarding management, in-hospital outcomes, short-term follow-up outcomes, and functional status of post-AMI patients requiring mechanical ventilator support so that potential improvements could be identified to optimize the utilization of healthcare resources. Prolonged ventilation is associated with an increased rate of adverse outcomes and an increased cost of management [7]; therefore, the identification of factors associated with prolonged ventilation, in particular, was a secondary outcome of interest in this audit, along with other adverse outcomes associated with intensive care including in-hospital bleeding rates, re-intubation, need for renal replacement therapy (RRT), and bed sores. Finally, the assessment of 180-day mortality and identification of factors associated with an increased risk of 180-day mortality were the objectives of this clinical audit.

## Materials and methods

### Study population and setting

This clinical audit was conducted at the critical care unit (CCU) of the NICVD, Karachi, Pakistan, from August 2021 to January 2022. All patients were managed as per a standard ICU protocol under the supervision of a team of consulting intensivists and cardiologists. This study was approved by the ethical review board of the NICVD (ERC-74/2021), and consent for inclusion in the study was taken from the patient's attendant or next of kin.

### Selection criteria

Inclusion criteria for the audit were adult patients (age >18 years) presenting with AMI who developed complications requiring mechanical ventilator support within 24 hours of initial symptom onset. The use of IABP was at the discretion of the physician.

Patients who did not undergo revascularization and showed signs of severe hypoxic brain injury (Glasgow Coma Scale (GCS) 3–8) were excluded from the study. Patients <18 years of

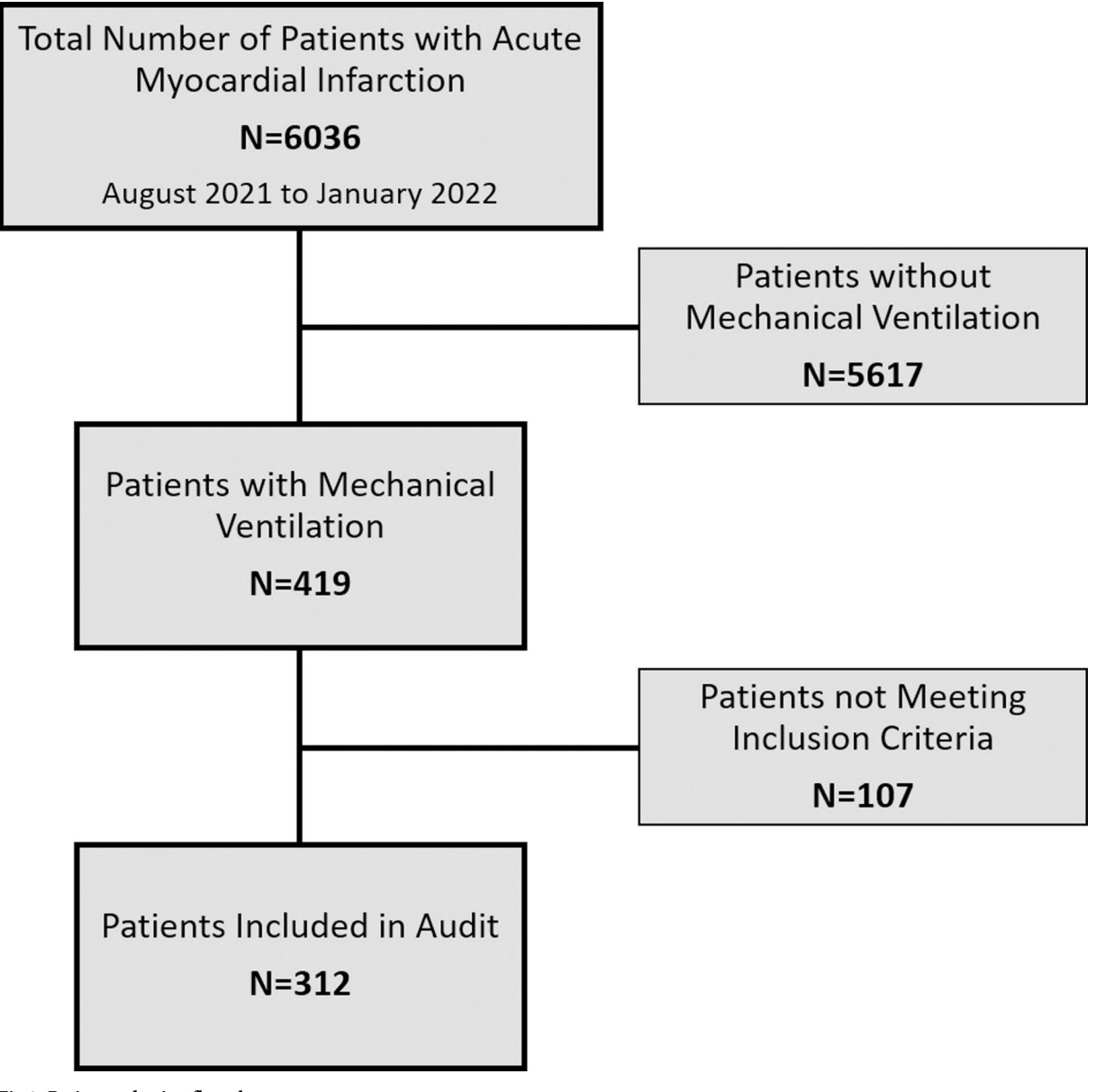

**Fig 1. Patients selection flow chart.**

age and patients for which consent could not be obtained were also excluded. The patient selection flow chart is provided in Fig 1.

### Study variables

The AMI was categorized in accordance with the fourth universal definition of myocardial infarction based on presentation electrocardiogram, history of typical chest pain, and rise and fall of cardiac enzymes [8]. All patients were managed by experienced consultant cardiologists as per the standard clinical practice and institutional protocols. All patients were pre-medicated with unfractionated heparin and dual antiplatelet therapy (DAPT) along with a bolus dose of glycoprotein inhibitors (IIb/IIIa).

Data for this clinical audit were prospectively collected using a structured proforma. The audit proforma consists of questions related to the patient's demographic characteristics and in-hospital metrics regarding management, disease course, outcome, and follow-up. Demographic variables included gender and age; the clinical characteristics consisted of data regarding duration from symptom onset to reperfusion, Killip class at presentation, type of MI and reperfusion strategy, angiographic findings (TIMI (thrombolysis in myocardial infarction) flow, culprit, or diseased vessels), co-morbid conditions, echocardiographic findings, intubation status (CPR (cardio-pulmonary resuscitation), time to achieve ROSC (return of spontaneous circulation) and elective intubation), and SOFA (Sequential Organ Failure Assessment) score. Detailed data of the use of any temporary pacemaker (TPM), IABP, vasopressors/inotropes, steroids, bicarbonate, antibiotics, laboratory markers, arterial blood gas (ABGs) and venous blood gas (VBGs), and fluid balance were also collected. Post-procedure transthoracic echocardiography was performed by an expert cardiologist using a 4 to 7 MHz-frequency probe for a 2D echocardiogram with color Doppler and spectral Doppler studies. Additionally, departmental echocardiography assessments were performed based on the clinical deterioration status of the patients.

The left ventricular end-diastolic pressure (LVEDP, mmHg) was assessed using left heart catheterization at admission and echocardiography at discharge. For the assessment of LVEDP based on echocardiography, in conjunction with a mitral peak early filling velocity E, the ratio of E/è was used to estimate left ventricular filling pressure (LVFP) as recommended by the American Society of Echocardiography (ASE) and European Society of Cardiology (ESC) for evaluating diastolic dysfunction in patients of heart failure with preserved ejection fraction (HFpEF) [9].

The use of steroids was subjective to the physicians' assessment of the patient's condition, which included; worsening of shock, increasing dosage and the number of vasopressors, suspicion of septic shock, and asthma/chronic obstructive pulmonary disease (COPD) exacerbation.

After 24 hours of invasive mechanical ventilation, patients were evaluated daily. Liberation from the ventilator was considered based on patients' hemodynamics, metabolic profile, ventilator requirements, and clinical condition [10]. Specific criteria under consideration were divided into clinical and laboratory parameters. Clinical criteria included the absence of arrhythmias, declining vasopressors (Noradrenaline dose of $\leq 0.1$ μg/kg/min) and less secretions, atelectasis, bronchospasm with a normal cough and gag reflex, and a GCS $\geq 10$. Lab parameters included improving hypoxemia defined as oxygen saturation (SaO2) of $\geq 90\%$ on fraction inspired oxygen (FiO2) $\leq 40\%$, normal electrolytes, hemoglobin level $\geq 8$ g/dl, RSBI (rapid shallow breathing index) $< 105$, and absence of any respiratory infection. The liberation protocol was based on gradually reducing pressure-support ventilation mode (PSV) to obtain an expiratory tidal volume of 8 ml/kg [10].

Data regarding the patient's hospital course was also recorded, which included duration of mechanical ventilation (hours), need for renal replacement therapy (RRT), any major bleeding event that required blood transfusion, and survival status. RRT need was determined by the treating physician based on the standard acute dialysis indications such as metabolic acidosis, refractory hyperkalemia, volume overload, or uremia causing encephalopathy and gastritis. Patients were subsequently followed at 30, 90, and 180 days for data on mortality and readmission to the hospital.

All patients in the study were also evaluated for frailty using the clinical frailty score elucidated by Dodson JA et al. [11], which was calculated using a combination of the patient's cognitive status and their ability to walk and perform activities of daily living. Frailty status was categorized as "no frailty (score: 0)", "mild frailty (score: 1–2)", and "Moderate to severe

(score≥3)". Frailty status was assessed at baseline (before the acute event), admission (CCU), discharge, and follow-up interval.

In accordance with the SCCM (Society of Critical Care Medicine), all discharged patients were enrolled in cardiac rehabilitation services, which included early mobilization of patients, sedation vacation, early liberation from ventilation, and early passive and active physiotherapy [12].

## Statistical analysis

The minimum required sample size of 265 for the audit was determined using a 45% expected rate of prolonged (>24 hours) mechanical ventilation, 6% absolute precision, and 95% confidence level; hence, this audit was conducted for the duration of 6 months. Collected data were analyzed using the statistical package for the social sciences (SPSS version 21. Patients were stratified into two groups based on the duration of mechanical ventilation of ≤ 24 hours or >24 hours (prolonged). Data were summarized as mean ± standard deviation (SD) or median [interquartile range (IQR)] for continuous variables and frequency and percentages for categorical variables. Two groups were compared with the help of appropriate independent sample t-test/Mann-Whitney test or Chi-square test to identify factors associated with prolonged ventilation. Repeated measures ANOVA (analysis of variance) was conducted to assess the improvement in frailty score over the follow-up period. All statistical tests were conducted at the 0.05 level of significance. Univariate and multivariable binary logistic regression analysis was performed to identify the predictors of prolonged MV duration and odds ratio (OR), and corresponding 95% confidence intervals (CI) were reported. Similarly, univariate and multivariable Cox regression analysis was performed to identify the predictors of 180-day mortality and hazard ratio (HR) along with the corresponding 95% CI are reported. The last known status of the patient was considered for the analysis. Variables with p-value <0.20 in univariate analysis were considered in the multivariable model. The dataset used for the analysis is provided as S1 Dataset.

## Results

A total of 312 consecutive patients with AMI requiring mechanical ventilator support were included in the study. The patients were predominantly male (72.8%) with a mean age of 60.3 ± 11.5 years, ranging from 25 to 96 years. The most prevalent co-morbid conditions were hypertension and diabetes mellitus, with an incidence of 82.4% and 34.9%, respectively. 28.2% of the patients were identified as smokers. A total of 91% (284) underwent PCI, 6.7% (21) underwent plain old balloon angioplasty (POBA) as the lesion was not suitable for stenting, while the remaining 2.2% (7) were advised either medical management or coronary artery bypass grafting (CABG) due to high-risk anatomy of the vessels.

The median duration of ventilation was 24 [IQR: 24–48] hours, ranging from a minimum of 12 to a maximum of 216 hours, with 51.3% requiring ventilator support of ≤ 24 hours. Prolonged (>24 hours) ventilator support was found to be associated with female gender (32.9% vs. 21.9%; p = 0.029), inferior wall myocardial infarction (MI) with right ventricular (RV) infarct (24.3% vs. 13.1%), sub-optimal (<III) TIMI flow (20.4% vs. 11.3%), culprit left main (7.9% vs. 2.5%), culprit right coronary artery (31.6% vs. 16.3%), diabetes (42.8% vs. 27.5%; p = 0.005), and obesity (7.2% vs. 1.3%; p = 0.008) when compared to ventilator duration of <24 hours, respectively (Table 1).

### Echocardiographic and electrocardiographic assessment

Post-MI echocardiography demonstrated mitral regurgitation (MR) in 52.2% of the patients, of which 33.1% (54) had mild-to-moderate MR. The mean ejection fraction was 32 ± 8%, with

**Table 1. Distribution of demographic, clinical, and angiographic characteristics and co-morbid conditions stratified by duration of mechanical ventilation.**

| | Total | Mechanical ventilation Duration | | P-value |
|---|---|---|---|---|
| | | ≤ 24 hours | > 24 hours | |
| **Total (N)** | **312** | **51.3% (160)** | **48.7% (152)** | |
| **Sex** | | | | |
| Male | 72.8% (227) | 78.1% (125) | 67.1% (102) | 0.029 |
| Female | 27.2% (85) | 21.9% (35) | 32.9% (50) | |
| **Age (years)** | 60.3 ± 11.5 | 60.1 ± 11.1 | 60.5 ± 11.9 | 0.748 |
| **Type of myocardial infarction (MI)** | | | | |
| Anterior wall MI (AWMI) | 63.1% (197) | 69.4% (111) | 56.6% (86) | 0.123 |
| AWMI + Right Bundle Branch Block | 1.3% (4) | 0.6% (1) | 2% (3) | |
| Inferior wall MI (IWMI) | 7.7% (24) | 8.1% (13) | 7.2% (11) | |
| IWMI + Right Ventricular Infarct | 18.6% (58) | 13.1% (21) | 24.3% (37) | |
| Posterior wall MI (PWMI) | 1.9% (6) | 2.5% (4) | 1.3% (2) | |
| Inferior posterior wall MI | 2.9% (9) | 1.9% (3) | 3.9% (6) | |
| Lateral wall MI | 1.9% (6) | 2.5% (4) | 1.3% (2) | |
| NSTEMI | 2.6% (8) | 1.9% (3) | 3.3% (5) | |
| **Reperfusion** | | | | |
| Percutaneous coronary intervention | 91% (284) | 91.9% (147) | 90.1% (137) | 0.269 |
| Plain old balloon angioplasty | 6.7% (21) | 5% (8) | 8.6% (13) | |
| Left heart catheterization | 2.2% (7) | 3.1% (5) | 1.3% (2) | |
| **Time to reperfusion (hours)** | 12 [7–24] | 10 [6–24] | 12 [8–36] | 0.080 |
| **Thrombolysis in myocardial infarction (TIMI) flow** | | | | |
| 0 | 0.6% (2) | 0.6% (1) | 0.7% (1) | 0.154 |
| I | 2.2% (7) | 1.3% (2) | 3.3% (5) | |
| II | 12.8% (40) | 9.4% (15) | 16.4% (25) | |
| III | 84.3% (263) | 88.8% (142) | 79.6% (121) | |
| **Cuprite vessel** | | | | |
| Left main | 5.1% (16) | 2.5% (4) | 7.9% (12) | 0.002 |
| Left anterior descending artery | 61.9% (193) | 70.6% (113) | 52.6% (80) | |
| Right coronary artery | 23.7% (74) | 16.3% (26) | 31.6% (48) | |
| Left circumflex artery | 7.4% (23) | 9.4% (15) | 5.3% (8) | |
| Obtuse marginal | 1% (3) | 0.6% (1) | 1.3% (2) | |
| Diagonal | 1% (3) | 0.6% (1) | 1.3% (2) | |
| **Other diseased vessels** | | | | |
| Left main | 14.1% (44) | 13.1% (21) | 15.1% (23) | 0.611 |
| Left anterior descending artery | 88.8% (277) | 90% (144) | 87.5% (133) | 0.484 |
| Right coronary artery | 68.6% (214) | 63.1% (101) | 74.3% (113) | 0.033 |
| Left circumflex artery | 57.1% (178) | 55.6% (89) | 58.6% (89) | 0.602 |
| Obtuse marginal | 3.8% (12) | 5.6% (9) | 2% (3) | 0.094 |
| Diagonal | 2.2% (7) | 1.3% (2) | 3.3% (5) | 0.224 |
| **Co-morbid conditions** | | | | |
| Hypertension | 82.4% (257) | 82.5% (132) | 82.2% (125) | 0.951 |
| Diabetes | 34.9% (109) | 27.5% (44) | 42.8% (65) | 0.005 |
| Smoker | 28.2% (88) | 26.9% (43) | 29.6% (45) | 0.592 |
| Chronic kidney disease | 7.1% (22) | 6.9% (11) | 7.2% (11) | 0.901 |
| Cerebrovascular accident | 4.5% (14) | 3.8% (6) | 5.3% (8) | 0.519 |
| COPD/Asthma | 3.8% (12) | 2.5% (4) | 5.3% (8) | 0.205 |
| Obese | 4.2% (13) | 1.3% (2) | 7.2% (11) | 0.008 |

(*Continued*)

**Table 1.** (Continued)

| | Total | Mechanical ventilation Duration | | P-value |
| --- | --- | --- | --- | --- |
| | | ≤ 24 hours | > 24 hours | |
| Chronic liver disease | 0.3% (1) | 0% (0) | 0.7% (1) | 0.304 |
| Congestive heart failure | 2.6% (8) | 1.3% (2) | 3.9% (6) | 0.132 |
| Ischemic heart diseases | 16% (50) | 14.4% (23) | 17.8% (27) | 0.415 |

NSTEMI: non ST elevation myocardial infarction, COPD: chronic obstructive pulmonary disease

left ventricular (LV) dysfunction in nearly all cases (96.2%). The mean LVEDP (left ventricular end-diastolic pressure) at the time of the procedure was 26.3 ± 9.0 mmHg. Right ventricular (RV) dysfunction and bi-ventricular failure were observed in 36.2% and 35.6% of the patients. Arrhythmias were observed in 30.1% of the patients, among which ventricular tachycardia (VT) (44.7%) and complete heart block (CHB) (43.6%) were the common types of arrhythmias.

Prolonged ventilator duration (>24 hours) was observed to be associated with RV dysfunction (47.4% vs. 25.6%; p<0.001), biventricular failure (46.7% vs. 25%; p<0.001), and elevated LVEDP (27.8 ± 9.4 mmHg vs. 25.0 ± 8.5 mmHg; p = 0.008) when compared to ventilator duration of <24 hours, respectively (Table 2). An LVEDP of ≥ 25 mmHg has a sensitivity of 63.1% and 55.1% of specificity for prolonged ventilation.

## Risk assessment and management

Mean SOFA score at admission was 6.8 ± 2.6, with 10.6% of patients presenting with a score of 9–11 and 3.2% with a score >11. The majority (80.4%) were electively, and 20.8% of patients required CPR with a mean time to ROSC of 5.2 ± 4.6 minutes. The mean duration of admission in the CCU was 3.0 ± 2.1 days; 6.7% of patients needed RRT, 17.6% required IABP placement, and 16.7% required placement of a TPM. Vasopressors/inotropes were needed in 63.8% of the patients, with a median duration of 24 [IQR: 24–48] hours ranging from a minimum of 1 to 144 hours. Steroids and bicarbonate were used in 29.8% and 63.5% of the patients, respectively. Median PCT (Procalcitonin) and CRP (C-reactive proteins) were 3 [IQR: 0.5–7] and 7 [IQR 2–13], and antibiotics were used in 14.4% of the patients. Overall 7.4% of the patients were re-intubated.

Prolonged ventilation time was found to be associated with a high SOFA score (7.9 ± 2.2 vs. 5.7 ± 2.4; p<0.001), PCT (4 [IQR: 1–10] vs. 1 [IQR: 0.335–3]; p = 0.001), CRP (10 [IQR: 3–18] vs. 5 [IQR: 2–10]; p = 0.001), positive balance (800 [IQR: 391–1500] vs. 500 [IQR: 300–1000]; p = 0.013), and negative balance (850 [IQR:1500–500] vs. 600 [IQR: 800–500]; p = 0.017), respectively. Vasopressors/Inotropes use (84.2% vs. 44.4%; p<0.001) and duration (48 [IQR: 24–48] hours vs. 24 [IQR 24–24] hours; p = 0.001) was higher among patients with prolonged ventilation duration, respectively. Similarly, there was a greater frequency of need for IABP (25.7% vs. 10%; p<0.001), steroids (40.8% vs. 19.4%; p<0.001), and re-intubation (11.8% vs. 3.1%; p<0.001) in patients with a prolonged ventilation duration (Table 3).

## Assessment of factors associated with prolonged MV duration

The binary logistic regression analysis for prolonged MV duration is presented in Table 4. On univariate analysis, female (1.75 [1.06–2.9]), inferior wall myocardial infarction with infarct (2.13 [1.18–3.84]), post-procedure TIMI flow grade <III (2.02 [1.08–3.79]), diabetes (1.97 [1.23–3.16]), obesity (6.16 [1.34–28.28]), right ventricular dysfunction (2.61 [1.62–4.21]),

**Table 2. Distribution of echocardiographic findings and arrhythmias stratified by duration of mechanical ventilation.**

| | Total | MV Duration | | P-value |
|---|---|---|---|---|
| | | ≤ 24 hours | > 24 hours | |
| **Total (N)** | **312** | **51.3% (160)** | **48.7% (152)** | |
| **Mitral regurgitation** | 52.2% (163) | 51.3% (82) | 53.3% (81) | 0.718 |
| Mild | 66.9% (109) | 70.7% (58) | 63% (51) | 0.572 |
| Moderate | 28.8% (47) | 25.6% (21) | 32.1% (26) | |
| Severe | 4.3% (7) | 3.7% (3) | 4.9% (4) | |
| **Ejection fraction (%)** | 32 ± 8 | 32 ± 7 | 31 ± 9 | 0.285 |
| **Ventricular septal rupture (VSR)** | 0.6% (2) | 0% (0) | 1.3% (2) | 0.145 |
| **Left ventricular (LV) dimensions** | | | | |
| Systolic (mm) | 34 ± 6 | 34 ± 6 | 34 ± 7 | 0.364 |
| Diastolic(mm) | 47 ± 6 | 47 ± 5 | 46 ± 6 | 0.207 |
| **LV dysfunction** | 96.2% (300) | 95% (152) | 97.4% (148) | 0.277 |
| **Right ventricular (RV) dimension (mm)** | 20.2 ± 2.6 | 20.0 ± 2.1 | 20.5 ± 3.1 | 0.112 |
| **TAPSE(mm)** | 16.6 ± 3.2 | 17.3 ± 2.7 | 15.9 ± 3.4 | <0.001 |
| **RV dysfunction** | 36.2% (113) | 25.6% (41) | 47.4% (72) | <0.001 |
| **biventricular failure** | 35.6% (111) | 25% (40) | 46.7% (71) | <0.001 |
| **LVEDP: at procedure (mmHg)** | 26.3 ± 9.1 | 25.0 ± 8.5 | 27.8 ± 9.4 | 0.008 |
| **LVEDP: at discharge (mmHg)** | 15.1 ± 5.4 | 14.4 ± 5.4 | 15.9 ± 5.3 | 0.020 |
| **Other echocardiographic findings** | | | | |
| Moderate to severe TR | 1.3% (4) | 1.3% (2) | 1.3% (2) | >0.999 |
| Moderate aortic regurgitation | 0.3% (1) | 0.6% (1) | 0% (0) | - |
| Severe aortic stenosis | 0.3% (1) | 0.6% (1) | 0% (0) | - |
| Mild to moderate PE | 1% (3) | 0% (0) | 2% (3) | - |
| LV thrombus | 1.9% (6) | 0.6% (1) | 3.3% (5) | 0.087 |
| **Arrhythmias** | 30.1% (94) | 23.8% (38) | 36.8% (56) | 0.695 |
| Complete heart block | 43.6% (41) | 42.1% (16) | 44.6% (25) | 0.501 |
| Ventricular tachycardia | 44.7% (42) | 44.7% (17) | 44.6% (25) | 0.067 |
| Atrial fibrillation | 4.3% (4) | 0% (0) | 7.1% (4) | 0.068 |
| Ventricular fibrillation | 5.3% (5) | 2.6% (1) | 7.1% (4) | 0.107 |
| Bradycardia | 6.4% (6) | 7.9% (3) | 5.4% (3) | 0.076 |
| Tachy-brady | 2.1% (2) | 0% (0) | 3.6% (2) | 0.316 |
| Idioventricular rhythm | 2.1% (2) | 0% (0) | 3.6% (2) | 0.316 |

TAPSE: tricuspid annular plane systolic excursion, LVEDP: left ventricular end-diastolic pressure, TR: tricuspid regurgitation, PE: pulmonary edema

biventricular failure (2.63 [1.63–4.25]), left ventricular end-diastolic pressure at the time of the procedure (1.03 [1.01–1.06]), arrhythmias (1.87 [1.15–3.06]), need of vasopressors/inotropes (6.69 [3.91–11.43]), temporary pacemaker placement (3.48 [1.8–6.73]), IABP placement (3.11 [1.65–5.84]), post-CPR intubation (1.72 [0.96–3.08]), SOFA score at admission (1.49 [1.33–1.67]), steroids used (2.87 [1.72–4.77]), bicarbonate used (2.71 [1.68–4.39]), CRP (1.06 [1.02–1.09]), albumin (0.32 [0.19–0.55]), and frailty score at admission (2.41 [1.47–3.94]) were found to be factors associated with prolonged MV duration. However, on multivariable analysis, albumin level was found to be the only independent predictor of prolonged MV duration with an adjusted OR of 0.42 [95% CI: 0.22–0.82; p = 0.011].

**Table 3. Risk assessment and management of patients in the critical care unit (CCU) stratified by duration of mechanical ventilation.**

| | Total | MV Duration | | P-value |
|---|---|---|---|---|
| | | ≤ 24 hours | > 24 hours | |
| **Total (N)** | **312** | **51.3% (160)** | **48.7% (152)** | |
| **Emergency intubation** | 80.4% (251) | 84.4% (135) | 76.3% (116) | 0.238 |
| **Post cardiac arrest** | 20.8% (65) | 17.5% (28) | 24.3% (37) | 0.282 |
| Time to achieve ROSC (min) | 5.2 ± 4.6 | 4.3 ± 4.0 | 5.8 ± 5.0 | 0.211 |
| **SOFA Score at admission** | 6.8 ± 2.6 | 5.7 ± 2.4 | 7.9 ± 2.2 | <0.001 |
| <9 | 86.2% (269) | 91.3% (146) | 80.9% (123) | 0.020 |
| 9 to 11 | 10.6% (33) | 7.5% (12) | 13.8% (21) | |
| >11 | 3.2% (10) | 1.3% (2) | 5.3% (8) | |
| NA | 0% (0) | 0% (0) | 0% (0) | |
| **SOFA Score at discharge** | 3.8 ± 4.9 | 2.3 ± 3.4 | 5.4 ± 5.6 | <0.001 |
| <9 | 84.3% (263) | 92.5% (148) | 75.7% (115) | 0.001 |
| 9 to 11 | 3.5% (11) | 2.5% (4) | 4.6% (7) | |
| >11 | 11.2% (35) | 4.4% (7) | 18.4% (28) | |
| NA | 1% (3) | 0.6% (1) | 1.3% (2) | |
| **Bedsores** | 6.4% (20) | 6.3% (10) | 6.6% (10) | 0.019 |
| 1 | 75% (15) | 80% (8) | 70% (7) | 0.020 |
| 2 | 25% (5) | 20% (2) | 30% (3) | |
| **Procalcitonin Concentration** (ng/mL) | 3 [0.5–7] | 1 [0.335–3] | 4 [1–10] | 0.001 |
| **C-reactive protein** (mg/L) | 7 [2–13] | 5 [2–10] | 10 [3–18] | 0.001 |
| **Albumin** (g/dL) | 3.6 ± 0.5 | 3.7 ± 0.4 | 3.4 ± 0.5 | <0.001 |
| **Cumulative Fluid Balance** | | | | |
| Negative | 39.5% (116) | 37.9% (58) | 41.1% (58) | 0.508 |
| Equal | 2.4% (7) | 3.3% (5) | 1.4% (2) | |
| Positive | 58.2% (171) | 58.8% (90) | 57.4% (81) | |
| **Positive fluid balance** (ml) | 509 [300–1000] | 500 [300–1000] | 800 [391–1500] | 0.013 |
| **Negative fluid balance** (ml) | 800 [1097–500] | 600 [800–500] | 850 [1500–500] | 0.017 |
| **Days in CCU** (days) | 3.02 ± 2.09 | 2.24 ± 0.97 | 3.85 ± 2.58 | <0.001 |
| **Need of RRT** | 6.7% (21) | 5% (8) | 8.6% (13) | 0.572 |
| Number of RRT sessions | 2.3 ± 0.7 | 2.2 ± 0.4 | 2.4 ± 0.9 | 0.937 |
| **FIO2 (%) at admission** | 60 [40–70] | 60 [40–80] | 50 [40–65] | 0.262 |
| **At admission arterial blood gas (ABGs)** | | | | |
| Po2 (mmHg) | 103 [76–149] | 104 [79–154] | 100 [72–139] | 0.138 |
| Co2 (mmHg) | 41 [31–51] | 43 [32–53] | 40 [29–49] | 0.107 |
| Sao2 (%) | 96 [92–98] | 97 [92–98] | 95 [92–98] | 0.011 |
| **At admission venous blood gas (VBGs)** | | | | |
| Po2 (mmHg) | 34 [30–40] | 35 [31–40] | 33 [30–40] | 0.224 |
| Co2 (mmHg) | 46 [40–58] | 47.5 [41–59] | 45 [40–56] | 0.219 |
| Svo2 (%) | 60 [53–68] | 61 [54–69] | 59 [52–68] | 0.149 |
| **FIO2 (%) at extubation** | 40 [40–40] | 40 [40–40] | 40 [40–40] | 0.012 |
| **At extubation arterial blood gas (ABGs)** | | | | |
| Po2 (mmHg) | 121 [95–158] | 120 [98–149] | 124 [94–171] | 0.799 |
| Co2 (mmHg) | 39 [34–44] | 39 [35–44] | 39 [33–45] | 0.885 |
| Sao2 (%) | 98 [96–99] | 98 [96–99] | 98 [96–99] | 0.340 |
| **At extubation venous blood gas (VBGs)** | | | | |
| Po2 (mmHg) | 37 [33–42] | 37.5 [33–41] | 36 [32–42] | 0.256 |
| Co2 (mmHg) | 47 [42–51] | 48 [43–51] | 47 [42–51] | 0.462 |

(*Continued*)

**Table 3.** (Continued)

| | Total | MV Duration | | P-value |
|---|---|---|---|---|
| | | ≤ 24 hours | > 24 hours | |
| Svo2 (%) | 63 [56–70] | 66 [59–70] | 62 [54–70] | 0.094 |
| **Oxygen extraction ratio** | | | | |
| At admission | 0.37 ± 0.12 | 0.37 ± 0.11 | 0.38 ± 0.12 | 0.659 |
| At Extubation | 0.37 ± 0.14 | 0.36 ± 0.14 | 0.38 ± 0.14 | 0.506 |
| **Vasopressors/Inotropes** | 63.8% (199) | 44.4% (71) | 84.2% (128) | <0.001 |
| Norepi (mic/kg/min) | 0.4 ± 0.6 | 0.4 ± 0.5 | 0.5 ± 0.6 | 0.104 |
| Epinephrine (mic/kg/min) | 1.5 ± 0.7 | 0 ± 0 | 1.5 ± 0.7 | - |
| Vasopressin (mic/kg/min) | 0.04 ± 0 | 0 ± 0 | 0.04 ± 0 | - |
| Phenylephrine (mic/kg/min) | 200 ± 0 | 0 ± 0 | 200 ± 0 | - |
| Dobutamine/milrinone (mic/kg/min) | 3 ± 1.2 | 3.8 ± 1.4 | 2.9 ± 1.1 | 0.174 |
| Dopamine (mic/kg/min) | 2.9 ± 1.0 | 3.8 ± 1.8 | 2.5 ± 0 | 0.178 |
| **Vasopressors/Inotropes Duration (hours)** | 24 [24–48] | 24 [24–24] | 48 [24–48] | 0.001 |
| **Temporary pacemaker** | 16.7% (52) | 8.8% (14) | 25% (38) | 0.064 |
| **Intra-aortic balloon pump** | 17.6% (55) | 10% (16) | 25.7% (39) | <0.001 |
| **Steroids** | 29.8% (93) | 19.4% (31) | 40.8% (62) | <0.001 |
| **Bicarbonate** | 63.5% (198) | 52.5% (84) | 75% (114) | 0.085 |
| **Reintubation** | 7.4% (23) | 3.1% (5) | 11.8% (18) | <0.001 |
| **Antibiotics** | 14.4% (45) | 7.5% (12) | 21.7% (33) | 0.320 |

CPR: cardiopulmonary resuscitation, SOFA: sequential organ failure assessment, RRT: renal replacement therapy, CCU: critical care unit, ROSC: return of spontaneous circulation

### Assessment of frailty and outcomes

The baseline mean frailty score (i.e., before the acute event) was 0.5 ± 1.0, with 74.7% of the patients being classified as 'no frailty.' The mean frailty score at admission increased to 3.7 ± 1.7, and a gradual improvement in the score was observed from discharge and at each follow-up interval, with mean values of 1.6 ± 1.5, 1.1 ± 1.6, 0.7 ± 1.2, and 0.5 ± 1.0 at discharge, 30-day, 90-day, and 180-day follow-up, respectively (repeated measures ANOVA p<0.001) (Fig 2).

The association between baseline frailty score and mechanical ventilation duration was insignificant, with a mean value of 0.6 ± 1.1 vs. 0.4 ± 1.0; p = 0.211 for the patients with and without prolonged (>24 hours) MV, respectively. However, the mean frailty score at discharge (1.3 ± 1.2 vs. 2.1 ± 1.7; p<0.001), 30-day (0.9 ± 1.4 vs. 1.4 ± 1.7; p = 0.025), 90-day (0.5 ± 1.1 vs. 1.0 ± 1.3; p = 0.013), and 180-day (0.4 ± 1 vs. 0.7 ± 1.1; p = 0.085) follow-up were higher for patients with MV duration of > 24 hours compared to patients with MV duration of ≤ 24 hours, respectively (Fig 2).

Major bleeding events (during hospital stay) were observed in 3.8% (12) of the patients. The incidence of bleeding events was found to be associated with prolonged ventilation with an event rate of 5.9% vs. 1.9% (p = 0.040) for the patients with a duration of MV > 24 hours and ≤ 24 hours, respectively.

The readmission rate was found to be 14.7% (37), with a significant association with the duration of ventilation. The readmission rate was 18.9% vs. 11.7%; p<0.001 for the patients with duration of MV > 24 hours and ≤ 24 hours, respectively.

Patients with cardiogenic shock (those who required vasopressors) had an in-hospital mortality rate of 28.6% (57/199) and a cumulative 180-day mortality rate of 39.2% (78/199). Post-

**Table 4. Binary logistic regression analysis for the prolonged (>24 hours) ventilator duration.**

| | Univariate | | Multivariable | |
|---|---|---|---|---|
| | OR [95% CI] | P-value | OR [95% CI] | P-value |
| Female | 1.75 [1.06–2.9] | 0.030 | 1.42 [0.71–2.83] | 0.325 |
| Age (years) | 1.00 [0.98–1.02] | 0.747 | - | - |
| Inferior wall myocardial infarction with Infarct | 2.13 [1.18–3.84] | 0.012 | 1.32 [0.54–3.21] | 0.538 |
| Time to reperfusion from symptom onset | 1.00 [1.00–1.01] | 0.437 | - | - |
| Post-procedure TIMI (thrombolysis in myocardial infarction) flow grade <III | 2.02 [1.08–3.79] | 0.028 | 1.64 [0.7–3.87] | 0.256 |
| Hypertension | 0.98 [0.55–1.76] | 0.951 | - | - |
| Diabetes | 1.97 [1.23–3.16] | 0.005 | 1.69 [0.89–3.23] | 0.112 |
| Smoker | 1.14 [0.7–1.87] | 0.592 | - | - |
| Chronic kidney disease | 1.06 [0.44–2.51] | 0.901 | - | - |
| History of cerebrovascular accident | 1.43 [0.48–4.21] | 0.521 | - | - |
| Chronic obstructive pulmonary disease/asthma | 2.17 [0.64–7.35] | 0.215 | - | - |
| Obese | 6.16 [1.34–28.28] | 0.019 | 4.55 [0.85–24.33] | 0.077 |
| Moderate to severe mitral regurgitation | 1.09 [0.7–1.69] | 0.719 | - | - |
| Ejection fraction (%) | 0.98 [0.96–1.01] | 0.282 | - | - |
| Right ventricular dysfunction | 2.61 [1.62–4.21] | <0.001 | 0.65 [0.03–13.54] | 0.781 |
| Biventricular failure | 2.63 [1.63–4.25] | <0.001 | 2.7 [0.13–56.35] | 0.523 |
| Left ventricular end-diastolic pressure at the time of procedure | 1.03 [1.01–1.06] | 0.009 | 1.01 [0.97–1.05] | 0.589 |
| Left ventricular thrombus | 5.41 [0.62–46.83] | 0.125 | 5.54 [0.51–59.65] | 0.158 |
| Arrhythmias | 1.87 [1.15–3.06] | 0.012 | 0.74 [0.31–1.74] | 0.488 |
| Need of vasopressors/inotropes | 6.69 [3.91–11.43] | <0.001 | 2.35 [0.97–5.71] | 0.059 |
| Temporary pacemaker placement | 3.48 [1.80–6.73] | <0.001 | 1.84 [0.65–5.20] | 0.252 |
| Intra-aortic balloon pump placement | 3.11 [1.65–5.84] | <0.001 | 1.42 [0.61–3.34] | 0.418 |
| Emergency intubation | 0.6 [0.34–1.05] | 0.074 | 1.73 [0.12–24.28] | 0.683 |
| Post CPR (cardiopulmonary resuscitation) intubation | 1.72 [0.96–3.08] | 0.070 | 2.31 [0.16–32.92] | 0.538 |
| SOFA score at admission | 1.49 [1.33–1.67] | <0.001 | 1.16 [0.96–1.40] | 0.117 |
| Steroids used | 2.87 [1.72–4.77] | <0.001 | 0.91 [0.43–1.94] | 0.805 |
| Bicarbonate used | 2.71 [1.68–4.39] | <0.001 | 1.94 [0.93–4.06] | 0.078 |
| C-reactive protein | 1.06 [1.02–1.09] | <0.001 | 1.02 [0.99–1.05] | 0.174 |
| Albumin | 0.32 [0.19–0.55] | <0.001 | 0.42 [0.22–0.82] | 0.011 |
| Positive fluid balance | 0.89 [0.57–1.39] | 0.600 | - | - |
| Frailty score at admission | 2.41 [1.47–3.94] | <0.001 | 1.63 [0.82–3.26] | 0.166 |

OR = odds ratio, CI = confidence interval

CPR patients' in-hospital mortality rate was 27.7% (18/65), and the cumulative 180-day mortality rate was 36.9% (24/65). Among patients with an IABP, the in-hospital mortality rate was 41.8% (23/55), and the cumulative 180-day mortality rate was 52.7% (29/55). There were 16 post-CPR patients with cardiogenic shock requiring an IABP, of which 8 (50%) died within the hospital, and the 180-day cumulative mortality rate was 62.5% (10/16).

The total survival rate was 80.4% (251) at the time of hospital discharge, 74.7% (233) at 30 days, 71.2% (222) at 90 days, and 68.6% (214) at 90-day follow-up (Fig 2). The survival rate was lower for the patients with duration of MV of > 24 hours compared to the patients with duration of MV of ≤ 24 hours at the time of discharge (69.7% vs. 90.6%; p<0.001) as well as at 30-day (62.5% vs. 86.3%; p<0.001), 90-day (59.9% vs. 81.9%; p<0.001), and 90-day (56.6% vs. 80%; p<0.001) follow-up, respectively (Fig 3).

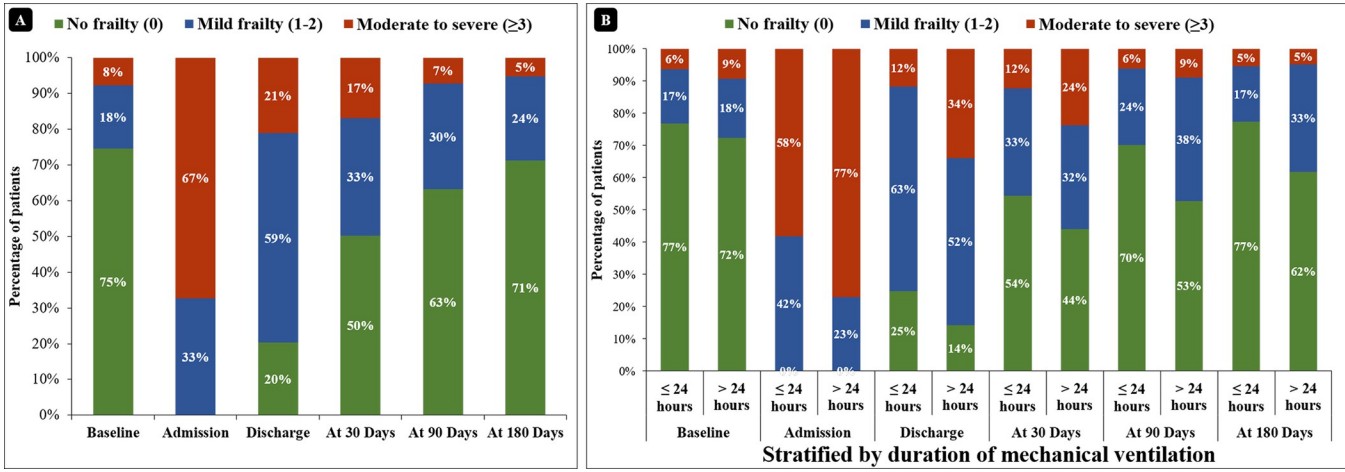

**Fig 2.** Distribution of frailty score at baseline and follow-up for total patients (A) and stratified by duration of mechanical ventilation (B).

## Assessment of factors associated with cumulative 180-day mortality

The Cox regression analysis for cumulative 180-day mortality is presented in Table 5. On multivariable analysis, Age (years), prolonged mechanical ventilator duration (>24 hours), and ejection fraction (%) were found to be the independent predictors of cumulative 180-day mortality with adjusted HR of 1.04 [95% CI: 1.02–1.07; p<0.001], 1.02 [95% CI: 1.01–1.03; p<0.001], and 0.95 [95% CI: 0.92–0.98; p = 0.003], respectively.

## Discussion

The purpose of this study was to track in-hospital outcomes, short-term follow-up outcomes, and functional status changes of AMI patients requiring mechanical ventilator support. In the audit of 312 patients, the majority (72.8%) were male, averaging 60.3 ± 11.5 years. The

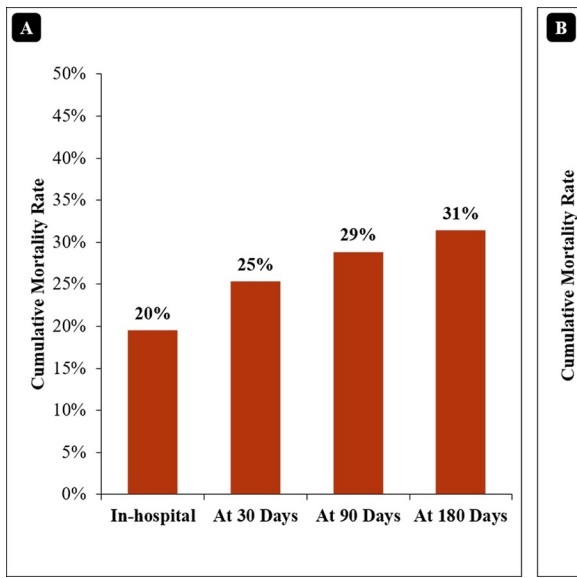

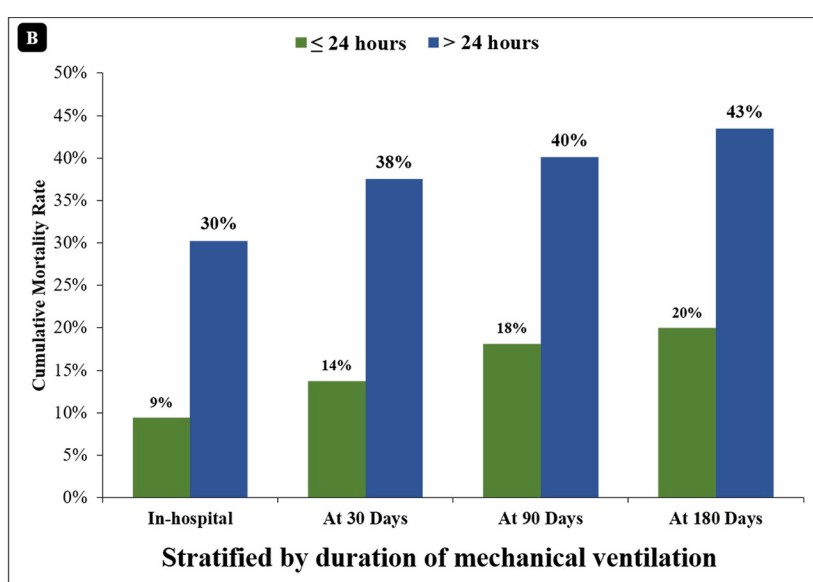

**Fig 3.** Distribution of cumulative in-hospital and follow-up mortality rate for total patients (A) and stratified by duration of mechanical ventilation (B).

**Table 5. Cox regression analysis for the 180-day cumulative mortality.**

| | Univariate | | Multivariable | |
|---|---|---|---|---|
| | HR [95% CI] | P-value | HR [95% CI] | P-value |
| Female | 1.30 [0.85–1.98] | 0.224 | - | - |
| Age (years) | 1.03 [1.01–1.05] | <0.001 | 1.04 [1.02–1.07] | <0.001 |
| Inferior wall myocardial infarction with Infarct | 1.41 [0.88–2.25] | 0.150 | - | - |
| Time to reperfusion from symptom onset | 1.00 [1.00–1.00] | 0.526 | - | - |
| Post-procedure TIMI (thrombolysis in myocardial infarction) flow grade <III | 2.00 [1.26–3.17] | 0.003 | 1.17 [0.64–2.13] | 0.607 |
| prolonged mechanical ventilator duration (>24 hours) | 1.01 [1.01–1.02] | <0.001 | 1.02 [1.01–1.03] | <0.001 |
| Hypertension | 0.95 [0.57–1.58] | 0.832 | - | - |
| Diabetes | 1.41 [0.94–2.11] | 0.092 | 1.15 [0.71–1.87] | 0.579 |
| Smoker | 1.00 [0.65–1.56] | 0.986 | - | - |
| Chronic kidney disease | 1.60 [0.83–3.07] | 0.160 | 1.13 [0.48–2.64] | 0.784 |
| History of cerebrovascular accident | 1.59 [0.74–3.43] | 0.238 | - | - |
| Chronic obstructive pulmonary disease/asthma | 1.45 [0.59–3.57] | 0.418 | - | - |
| Obese | 1.97 [0.91–4.25] | 0.084 | 1.88 [0.77–4.63] | 0.168 |
| Moderate to severe mitral regurgitation | 1.48 [0.98–2.22] | 0.059 | 1.33 [0.80–2.20] | 0.270 |
| Ejection fraction (%) | 0.93 [0.9–0.95] | <0.001 | 0.95 [0.92–0.98] | 0.003 |
| Ventricular septal rupture | 5.27 [1.29–21.52] | 0.021 | 1.46 [0.25–8.62] | 0.677 |
| Right ventricular dysfunction | 1.89 [1.27–2.8] | 0.002 | - | - |
| Biventricular failure | 1.95 [1.31–2.9] | <0.001 | 1.24 [0.75–2.04] | 0.403 |
| Left ventricular end-diastolic pressure at the time of procedure | 1.03 [1.00–1.05] | 0.023 | 1.00 [0.98–1.03] | 0.897 |
| Left ventricular thrombus | 1.07 [0.26–4.36] | 0.920 | - | - |
| Arrhythmias | 1.10 [0.72–1.68] | 0.660 | - | - |
| Need of vasopressors/inotropes | 2.50 [1.53–4.09] | <0.001 | 0.83 [0.37–1.86] | 0.652 |
| Temporary pacemaker placement | 1.60 [1.00–2.58] | 0.052 | 1.00 [0.55–1.81] | 0.991 |
| Intra-aortic balloon pump placement | 2.27 [1.47–3.5] | <0.001 | 1.30 [0.70–2.42] | 0.400 |
| Emergency intubation | 0.75 [0.47–1.19] | 0.225 | - | - |
| Post CPR (cardiopulmonary resuscitation) intubation | 1.28 [0.79–2.07] | 0.316 | - | - |
| SOFA score at admission | 1.27 [1.17–1.37] | <0.001 | 1.07 [0.92–1.26] | 0.370 |
| Steroids used | 2.26 [1.52–3.37] | <0.001 | 1.61 [0.93–2.78] | 0.088 |
| Bicarbonate used | 1.42 [0.92–2.2] | 0.111 | 0.66 [0.34–1.3] | 0.235 |
| C-reactive protein | 1.02 [1.00–1.03] | 0.008 | 1.00 [0.98–1.03] | 0.921 |
| Albumin | 0.58 [0.38–0.87] | 0.008 | 0.75 [0.45–1.24] | 0.258 |
| Positive fluid balance | 0.93 [0.62–1.38] | 0.712 | - | - |
| Frailty score at admission | 2.75 [1.61–4.71] | <0.001 | 1.21 [0.63–2.35] | 0.562 |

HR = hazard ratio, CI = confidence interval

predominant type of presentation was an anterior wall MI, presenting in 63.1%. Prolonged ischemia could be among the possible reasons for the need for mechanical ventilator support in these patients, as median ischemic duration was observed to be 12 [IQR: 7–24] hours. Additionally, hypertension was the most common co-morbid condition, followed by diabetes. Fewer patients had chronic kidney diseases (CKD) (7.1%) and COPD/Asthma (3.8%). MR was a common structural abnormality in these patients. Life-threatening arrhythmias, such as CHB and VT, were prevalent in more than 1/4th (30.1%) of the patients. The rate of emergency intubation was observed to be 80.4%, and around 1/5th (20.8%) were post-CPR patients with an average of 5.15 ± 4.64 minutes to achieve ROSC. IABP and TPM were placed in 17.6% and 16.7% of the patients. A median duration on a ventilator was observed to be 24 [IQR: 24–48]

hours, during which a majority (63.8%) needed inotropic/vasopressors support. The in-hospital survival rate was observed to be 80.4%, and the cumulative survival rate at subsequent follow-ups at 30-day, 90-day, and 180-day was observed to be 74.7%, 71.2%, and 68.6%, respectively. Readmission and bleeding events were noted in 14.7% and 3.8% of the patients, respectively. The need for renal replacement therapy was observed for 6.7%, and only 7.4% of the patients needed re-intubation.

Prolonged ventilator need was observed in nearly half (48.7%) of the patients, which was found to be associated with various factors such as female gender, inferior wall MI with RV infarct, sub-optimal (<III) TIMI, culprit left main (LM)/right coronary artery (RCA), diabetes, obesity, RV dysfunction, biventricular failure, elevated LVEDP, high admission SOFA score, PCT, CRP, positive balance, negative balance, and frailty status at admission. Prolonged ventilator was found to be associated with vasopressors/inotropes use and duration along with the use of IABP and steroids. Prolonged ventilator was also found to be associated with an increased risk of re-intubation, bleeding events, in-hospital mortality, and mortality at subsequent follow-ups at 30-day, 90-day, and 180-day.

Risk factors for cardiovascular diseases are highly prevalent in Pakistan, especially in urban areas [13]. With official GDP of 1600 US dollars, a lack of health awareness, and expensive medical treatment and evaluation, most people are without any medication or lifestyle modification, leading to a significant increase in early cardiovascular events.

In 2017–2018 with the support of the local government, the leadership of NICVD took the initiative of free of cost cardiovascular treatment, which led to a surge of patients with acute coronary syndrome and patients not only from the city of Karachi or adjoining areas of the province of Sindh but from all over the country which put a remarkable load on the existing infrastructure of the institute as well as the financial cost. Since 2018 the development of the department of critical care and the development of a training program with adequately trained faculty, the idea was to evaluate the department's performance and the outcome of this unique population.

Compared to an audit published by Irish Heart Attack Audit National Report [14], STEMI (ST-segment elevation myocardial infarction) was diagnosed in 85% of the patients admitted to their Cardiac hospital, with comparable demographic findings to ours (78% male; median age 61 years). The most common co-morbid conditions reported were hypertension and hypercholesterolemia; we found hypertension followed by diabetes to be the most common co-morbid.

In this study, hospital ICU mortality is 19.6% for a survival rate of 80.4% which is in contrast to what has been reported by K Parhar et al. in their study published in 2018, which reported an ICU mortality of 33.7%. This could be attributed to a difference in the patient population, which included 36.6% and 31.7% of patients being admitted to the ICU following an out-of-hospital and intra-hospital cardiac arrest, compared to only 20.8% of patients in our study requiring CPR [15]. Another study reports an in-hospital mortality rate of 25% in patients who needed both invasive mechanical ventilation and IABP with PCI, which is comparable to our overall mortality rate, even though not all patients required mechanical ventilator support. The same study reported higher mortality rates of 66% for patients who required PCI with invasive mechanical ventilation and partial IABP [16]. Another study published showed a mortality rate of 37.3% at 30 days and 40.0% at 90 days in patients requiring PCI complicated by cardiogenic shock requiring vasopressors and invasive mechanical ventilation and IABP if needed in the ICU [17]. This is higher than our mortality rate reported at 30 days and 90 days which is 25.3% and 28.8%, respectively. As lower post-PCI TIMI flow has been associated with poorer outcomes post-PCI, our lower mortality rates can be due to lower TIMI

0 and TIMI I flow scores of 0.6% and 2.2% as compared to 9.0% and 3.2%, respectively, in the previously quoted study [17].

Factor associated independently with higher ICU mortality was a higher SOFA mean 10.4 score in non-survivors (odd ratio for increased mortality with every point increase in SOFA being 1.43 [1.20–1.70 CI] in the study by K Parher and team [15], which is somewhat similar to our findings of patients with higher SOFA score requiring prolonged ventilation > 24 hours and thus having higher mortality. Initially, the SOFA score was described as a sepsis-related organ failure assessment tool, the utility of the score for the assessment of acute morbidity in a range of critical illnesses was recognized early, and the title changed. Nowadays, the SOFA score is used to assess organ dysfunction in non-septic critical states and is a reasonable predictive tool in ICU patient outcomes [18].

A 2013 study reported a mean duration of invasive mechanical ventilation of 2 days. It showed significantly higher mortality, comparable with our findings of increased in-hospital and 30, 90, and 180 days mortality in patients requiring invasive mechanical ventilation >24 hours. Also, the patients requiring mechanical ventilation showed poorer left ventricle function, unfavorable hemodynamics, and unfavorable hemodynamics, suggesting impending or established shock [3]. Our study showed almost comparable results except for a higher incidence of RV failure, with lower TAPSE and biventricular failure being more associated with patients requiring prolonged ventilation. Higher requirements of vasopressors with higher SOFA scores suggesting more severe organ failure were also comparable with the previous study. Fever and raised CRP are common in post-MI patients. Therefore our patients are thoroughly assessed for the cause of shock using point-of-care Ultrasound and PCT. Most of the patients in our study were in cardiogenic shock, which explains the high use of vasopressors [19].

There is still a scarcity of studies in the literature exploring factors associated with prolonged ventilation post-PCI; however, a prior study involving patients post-CABG surgery observed multiple clinical features as risk factors for prolonged ventilation >48 hours, specifically a history of heart failure or poor LV function, COPD, kidney failure with a creatinine >2.3mg/dl, hemoglobin < 12gm/dl and LVEF < 35% in the pre-surgical period and higher blood transfusions, presence of persistent shock, use of IABP and vasopressors and venous oxygen saturation of <60% in the post-surgical period [20]. This is consistent with our findings, although we reported higher numbers of vasopressor doses and duration and more need for IABP support in patients requiring prolonged invasive mechanical ventilation. However, they defined prolonged mechanical ventilation as ventilator use lasting > 48 hours, while we defined it as a duration > 24 hours. While the study's results were largely comparable to ours, it highlights the need for more research to establish a firmer definition.

Frailty represents decreased physiological reserve and serves as a better indicator of biological age. Frailty has been associated with increased all-cause mortality, risk of bleeding, and readmission in elderly ACS patients in a meta-analysis that included 15 studies [21]. Another meta-analysis reported an increase in mortality in frail patients after acute coronary syndrome; the study compared mortality numbers at initial presentation, six months after the events, and >12 months after the event [22]. A meta-analysis involving patients with coronary artery disease undergoing PCI also showed frailty as an independent risk factor for all-cause mortality and major cardiovascular events [23]. A similar recent meta-analysis by Tse G et al. reported frailty as a significant predictor of all-cause mortality following PCI in patients with ACS [24]. Our study didn't directly assess mortality according to frailty scores. Instead, it aimed to demonstrate that higher frailty scores were associated with prolonged ventilation, contributing to higher mortality.

The research findings provide valuable clinical insights for managing post-AMI patients, especially those requiring prolonged mechanical ventilation. By recognizing risk factors early, developing tailored treatment plans, addressing potential complications, and providing long-

term follow-up, healthcare providers can improve patient outcomes and reduce cumulative mortality in this vulnerable population. Additionally, the identification of admission albumin levels as an independent predictor of prolonged MV duration opens avenues for nutritional interventions and further research into this aspect of care.

One of the key limitations of our study is its single-center design, which may restrict the generalizability of our findings to a broader population. Conducting the study in a single center could lead to potential biases related to the unique patient characteristics and healthcare practices within that specific institution. To enhance the applicability of our results, future studies should involve multiple centers and diverse patient populations to validate and extend the conclusions.

Another important limitation is the lack of control for certain confounding factors that could influence the relationship between prolonged mechanical ventilation and outcomes. Specifically, we did not account for the severity of the inciting event or the time to revascularization, which might have an impact on the association between prolonged mechanical ventilation and outcomes. Secondly, management strategy and or stage PCI for patients with multivessel disease has not been documented, and it may have an impact on post-discharge outcomes. The inclusion of these variables in the analysis could offer a more comprehensive understanding of the interplay between mechanical ventilation and outcomes, providing valuable insights into their complex relationship.

Furthermore, it is essential to acknowledge that the data utilized in this study was collected as part of an ongoing initiative aimed at improving resource utilization and management outcomes within our institution. As a result, the data analysis was conducted post-hoc, which may introduce certain limitations and potential biases in the interpretation of the results. To mitigate this concern, future studies should be designed prospectively, with a predefined research plan and well-defined endpoints, ensuring a more robust and unbiased investigation of the relationship between mechanical ventilation and frailty.

## Conclusion

The need for prolonged mechanical ventilator support in post-AMI patients is associated with a higher fatality rate and an increased risk of short-term mortality. Specialized care by a dedicated team of intensivists can improve the fate of the patients as well as the management outcomes of the healthcare system. In the current study, relatively better survival and short-term recovery have been witnessed in otherwise high-risk subgroups of patients. However, prolonged ventilator duration has significant prognostic implications; hence, tailored early recognition of high-risk patients needing more aggressive care can improve the outcomes. In the search for superlative derivatives and factors, a potential role of frailty assessment for risk stratification has been identified in addition to the demographic and clinical covariates. Hence, further studies are warranted to identify significant clinical determinants of prolonged ventilator duration and adverse clinical outcomes for these patients.

## Supporting information

**S1 Dataset.**
(XLSX)

## Acknowledgments

The authors wish to acknowledge the support of the Clinical Research Department staff members of the National Institute of Cardiovascular Diseases (NICVD), Karachi, Pakistan.

## Author Contributions

**Conceptualization:** Muhammad Imran Ansari, Madiha Umair, Musa Karim.

**Data curation:** Madiha Umair, Lalarukh Taimoor, Zohaib Abubaker, Jawed Abubaker.

**Formal analysis:** Muhammad Imran Ansari, Musa Karim.

**Investigation:** Madiha Umair, Aziz-ur Rehman Memon, Muhammad Sohaib Arif, Jawed Abubaker.

**Methodology:** Muhammad Imran Ansari, Zohaib Abubaker.

**Project administration:** Aziz-ur Rehman Memon, Muhammad Sohaib Arif.

**Resources:** Lalarukh Taimoor.

**Software:** Zohaib Abubaker, Musa Karim.

**Supervision:** Nadeem Qamar.

**Validation:** Aziz-ur Rehman Memon, Jawed Abubaker.

**Visualization:** Aziz-ur Rehman Memon, Muhammad Sohaib Arif.

**Writing – original draft:** Muhammad Imran Ansari, Musa Karim.

**Writing – review & editing:** Nadeem Qamar.

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
