## [Decision Letter · Decision Letter 0]

20 Jul 2023

PONE-D-23-18200Mechanical Ventilation in Acute Myocardial Infarction: Outcomes from a Prospective Audit at a Cardiovascular Hospital in PakistanPLOS ONE

Dear Dr. Ansari,

Thank you for submitting your manuscript to PLOS ONE. After careful consideration, we feel that it has merit but does not fully meet PLOS ONE’s publication criteria as it currently stands. Therefore, we invite you to submit a revised version of the manuscript that addresses the points raised during the review process.

We look forward to receiving your revised manuscript.

Kind regards,

Redoy Ranjan, MBBS, MRCSEd, Ch.M., MS (CV&TS), FACS

Academic Editor

PLOS ONE

Journal Requirements:

3. We are unable to open your Supporting Information file [S1 Dataset: Dataset in SPSS format]. Please kindly revise as necessary and re-upload.

Reviewers' comments:

Reviewer's Responses to Questions

**Comments to the Author**

1. Is the manuscript technically sound, and do the data support the conclusions?

Reviewer #1: Yes

Reviewer #2: Yes

Reviewer #3: Yes

2. Has the statistical analysis been performed appropriately and rigorously? 

Reviewer #1: Yes

Reviewer #2: Yes

Reviewer #3: Yes

3. Have the authors made all data underlying the findings in their manuscript fully available?

Reviewer #1: Yes

Reviewer #2: No

Reviewer #3: No

4. Is the manuscript presented in an intelligible fashion and written in standard English?

Reviewer #1: Yes

Reviewer #2: No

Reviewer #3: Yes

5. Review Comments to the Author

Reviewer #1: We have made a review for Manuscript Number PONE-D-23-18200; entitled:

Mechanical Ventilation in Acute Myocardial Infarction: Outcomes from a Prospective Audit at a Cardiovascular Hospital in Pakistan. We have some comments:

1- A flow chart is needed.

2- The calculation of the minimal sample size needed to conduct the study should be demonstrated in the methodology.

3- The authors better to determine the types of myocardial infarction included, if they enrolled all MI types, then better to define MI in the methodology.

4- The authors should improve the coherence of the introduction. Frailty is an independent topic which might not be suitable to analyse here.

5- The discussion should be re written in a more academic manner. Start the first paragraph with the main findings you like highlight, then proper critical appraisal in the light of prior literatures in the subsequent paragraphs.

6- In a paragraph before the limitations, highlight the clinical implications of the study in a clear and practical manner.

7- All the potential limitations should be highlighted properly and in acceptable order in the limitations section.

8- Improve the quality and resolution of the figures.

9- Describe the abbreviations once first mentioned. In the abstract; IABP. In the text; GCS, RSBI, etc...

Regards

Reviewer #2: The importance for the readership is explicitly justified. One concrete aim are formulated

The literature search is described in details, including search terms and inclusion criteria.

Key statement are supported by references.

Appropriate evidence is generally presented

Reviewer #3: -Other diseased vessel are not clear if significant lesions and need further revascularization or not.

-Noninvasive mechanical ventilation still a good choice in acute cardiac insult if patient can cope and not mentioned in your study.

- no comments on mechanical complications as rupture chordae and papillary muscle dysfunction as a cause of MR .

- the need of glycoprotein IIB IIIA , enoxaparin.UFH are not mentioned as affecting the bleeding risk.

- The indication of RRT should be clarified (anuria, metabolic acidosis or volume overload)

6. PLOS authors have the option to publish the peer review history of their article (what does this mean?). If published, this will include your full peer review and any attached files.

Reviewer #1: **Yes: **Rami Riziq Yousef Abumuaileq

Reviewer #2: No

Reviewer #3: No

---

## [Author Response · Author response to Decision Letter 0]

26 Jul 2023

Dear Editor,

Thank you for sharing valuable feedback of reviewers regarding our submission to the PLOS ONE titled “Mechanical Ventilation in Acute Myocardial Infarction: Outcomes from a Prospective Audit at a Cardiovascular Hospital in Pakistan”. Manuscript ID: [PONE-D-23-18200] - [EMID:24ddb4a8281ed9f6]

We have revised the manuscript as per the specific comments as listed below. We believe these changes have strengthened the quality of our manuscript and that you will find it suitable for publication in the Journal.

Reviewer 1

Comment 1- A flow chart is needed.

Response: Thank you for the comment. As suggested, a patient selection flow chart is added to the manuscript.

Comment 2- The calculation of the minimal sample size needed to conduct the study should be demonstrated in the methodology.

Response: Thank you for the comment. As suggested, details regarding consideration for the minimum sample size are added in the methods section.

Comment 3- The authors better to determine the types of myocardial infarction included, if they enrolled all MI types, then better to define MI in the methodology.

Response: Thank you for the comment. As suggested, details regarding the categorization of MI are added to the methods section.

Comment 4- The authors should improve the coherence of the introduction. Frailty is an independent topic which might not be suitable to analyse here.

Response: Thank you for the comment. As suggested, frailty has been omitted from the introduction and this section has been improved for the flow.

Comment 5- The discussion should be re written in a more academic manner. Start the first paragraph with the main findings you like highlight, then proper critical appraisal in the light of prior literatures in the subsequent paragraphs.

Response: Thank you for the comment. As suggested, we have added study highlights in the first paragraph and a discussion of the findings with reference to the literature in subsequent paragraphs.

Comment 6- In a paragraph before the limitations, highlight the clinical implications of the study in a clear and practical manner.

Response: Thank you for the comment. As suggested, we have added the clinical implications of the study before the limitation section.

Comment 7- All the potential limitations should be highlighted properly and in acceptable order in the limitations section.

Response: Thank you for the comment. As suggested, we have elaborated on the limitation section.

Comment 8- Improve the quality and resolution of the figures.

Response: Thank you for the comment. As suggested we have improved the resolution of figures

Comment 9- Describe the abbreviations once first mentioned. In the abstract; IABP. In the text; GCS, RSBI, etc...

Response: Thank you for pointing it out. As suggested, we have added full forms of all the abbreviations at the first instance

Reviewer 2

Comment The importance for the readership is explicitly justified. One concrete aim are formulated

The literature search is described in details, including search terms and inclusion criteria.

Key statement are supported by references.

Appropriate evidence is generally presented

Response: Thank you for the kind appreciation and acknowledgment of the work

Reviewer 3

Comment 1- Other diseased vessel are not clear if significant lesions and need further revascularization or not.

Response: Thank you for the comment. However, the focus of the study was the audit of ICU care but stage PCI and other management strategies for other diseased vessels may have an impact on short-term outcomes, unfortunately, we had not documented this information in our audit and it has been acknowledged in the limitation section.

Comment 2-Noninvasive mechanical ventilation still a good choice in acute cardiac insult if patient can cope and not mentioned in your study.

Response: Thank you for the comment. However, our inclusion criteria were explicitly those patients who required invasive mechanical ventilation after AMI. 

Comment 3- no comments on mechanical complications as rupture chordae and papillary muscle dysfunction as a cause of MR .

Response: Thank you for pointing it out. We have document MR based on departmental ECHO but unfortunately, the cause of MR was not documented in this audit. 

Comment 4- the need of glycoprotein IIB IIIA , enoxaparin.UFH are not mentioned as affecting the bleeding risk.

Response: Thank you for pointing it out. All patients were managed as per the standard clinical practice guidelines. Details are added in the methods section.

Comment 5- The indication of RRT should be clarified (anuria, metabolic acidosis or volume overload

Response: Thank you for the comment. As suggested we have added indications for the RRT in our methods section.

---

## [Decision Letter · Decision Letter 1]

8 Aug 2023

Mechanical Ventilation in Acute Myocardial Infarction: Outcomes from a Prospective Audit at a Cardiovascular Hospital in Pakistan

PONE-D-23-18200R1

Dear Dr. Ansari,

We’re pleased to inform you that your manuscript has been judged scientifically suitable for publication and will be formally accepted for publication once it meets all outstanding technical requirements.

Kind regards,

Redoy Ranjan, MBBS, MRCSEd, Ch.M., MS (CV&TS), FACS

Academic Editor

PLOS ONE

---

## [Editor Report · Acceptance letter]

10 Aug 2023

PONE-D-23-18200R1 

Mechanical Ventilation in Acute Myocardial Infarction: Outcomes from a Prospective Audit at a Cardiovascular Hospital in Pakistan 

Dear Dr. Ansari:

I'm pleased to inform you that your manuscript has been deemed suitable for publication in PLOS ONE. Congratulations! Your manuscript is now with our production department. 

Kind regards, 

on behalf of

Dr. Redoy Ranjan 

Academic Editor

PLOS ONE